# Glucocorticoids intrinsically redirect naïve CD4+ T cells to the bone marrow for preservation in malnourished mice

Madeline L. Smith[1,§], Jacob Hanes[1,§], Takesha R. Foster[1,*,§], Clay Phillips[1], Kwesi A. Dadzie[1,‡] and Melanie R. Gubbels Bupp[¶]

## ABSTRACT

Malnutrition impairs immunity and contributes significantly to child mortality. Among other immune changes during malnutrition, effector T cells experience a decline in number and function. As little is known about the effect of malnutrition on naïve T cells, we examined the impact of malnutrition on the naïve T-cell population. During malnutrition, the number of naïve T cells decreased in the lymph nodes and increased in the bone marrow, where naïve CD4+ T cells experienced less apoptosis than controls. An adoptive transfer experiment revealed that malnourished naïve CD4+ T cells preferentially migrated to the bone marrow, while *ad libitum*-fed control cells preferentially migrated to the lymph nodes in a T-cell-intrinsic fashion. Additionally, T-cell-specific lack of the glucocorticoid receptor greatly reduced the number of naïve T cells in the bone marrow of malnourished mice, while dexamethasone treatment elevated the number of naïve CD4+ T cells in the bone marrow. T-cell sensitivity to glucocorticoids was required for elevated expression of CXCR4 and CCR7 in naïve CD4+ T cells during malnutrition. Overall, naïve CD4+ T-cell migration to the bone marrow during malnutrition is intrinsic, requires sensitivity to glucocorticoids, and likely contributes to naïve T-cell preservation in mice.

KEY WORDS: Malnutrition, Naïve T cells, Bone marrow, Glucocorticoids

## INTRODUCTION

Many illnesses are intensified by malnutrition's impairment of the immune system. For example, malnutrition is known to increase the risk of contracting tuberculosis as well as the disease's severity and mortality (Hood, 2013). Mortality is similarly elevated in malnourished (MAL) pneumonia patients and MAL children suffering from diarrhea (Kirolos et al., 2021; Tickell et al., 2020). On average, 56% of child deaths in developing countries result from the exacerbation of infection by malnutrition, and mild-to-moderate malnutrition accounts for over 80% of these deaths (Pelletier et al., 1995). Thus, malnutrition does not have to be severe to be fatal. Malnutrition can also impair the efficacy of vaccines; MAL infants who received oral polio vaccines had lower antibody levels than their well-nourished peers, and oral rotavirus vaccines may also be less effective in MAL children (Burnett et al., 2021; Saleem et al., 2015). An improved understanding of the cellular mechanisms behind malnutrition-related immune dysfunction could help to bolster vaccines for MAL individuals.

Malnutrition is associated with reductions in the number of immune cells and the reduced responsiveness of lymphocytes in humans (Rytter et al., 2014), laboratory animals (Takakuwa et al., 2019; Xu et al., 2017; Wing et al., 1988; Murphy et al., 2016), and captive animals, including invertebrates (Alonso-Alvarez and Tella, 2001; Bilbo and Nelson, 2004; Brzek and Konarzewski, 2007; Butt et al., 2007; Lochmiller et al., 1993; Martin et al., 2008; Merlo et al., 2016; Pascual et al., 2006; Xu et al., 2008). MAL children and laboratory rodents demonstrate reductions in the mass and cellularity of the spleen, thymus, and lymph nodes, with mostly proportional reductions in the total numbers of all immune subsets in those organs (Rytter et al., 2014; Takakuwa et al., 2019; Xu et al., 2017; Wing et al., 1988; Murphy et al., 2016). For example, monocytes and dendritic cells decrease in number during malnutrition in mice, and both B-cell and natural-killer-cell counts and function are reduced in MAL children and mice (Collins et al., 2019; Gerriets and MacIver, 2014; McGee and McMurray, 1988; Munteanu and Schwartz, 2022; Niiya et al., 2007; Noor et al., 2025). Thus, a variety of immune cells experience declines in population and function because of malnutrition. Malnutrition also negatively impacts T-cell number and function, though perhaps not as dramatically as other immune cell populations.

The effect of malnutrition on the number of T cells is influenced by age and the presence of concurrent infections. In the absence of infection, malnutrition appears to reduce peripheral T-cell counts. For example, young adult patients with restricting-type anorexia nervosa experienced reductions in the number of CD4+ and CD8+ T cells in the blood (Saito et al., 2007). In the context of infection, marasmus in adult patients did not affect the percentages of CD4+ or CD8+ T cells in the blood. However, adult patients with more severe kwashiorkor-like malnutrition and concurrent infections displayed reductions in the percentages of blood CD4+ and CD8+ T cells (Abbott et al., 1986). Studies undertaken with MAL children in the context of infection showed a lack of effect on the total number of T cells in the blood (Nájera et al., 2007; 2004). However, the number of CD4+ T cells was reduced in MAL children with infections compared to well-nourished children, and MAL children with infections also had lower percentages of effector T cells than well-nourished children with infections (Nájera et al., 2007, 2004). Additionally, elderly people with general protein-energy malnutrition did not show declines in total lymphocyte numbers (Kuzuya et al., 2005). Thus, isolating the effect of malnutrition alone on T-cell populations is difficult, as T-cell dynamics are likely influenced by patient age and the presence of infection. An additional limitation

[1]Department of Biology, Randolph-Macon College, Ashland, VA 23005, USA.
*Present address: Department of Neuroscience, University of Virginia, Charlottesville, VA, USA. ‡Present address: Alexandria, VA, USA.
§These authors contributed equally to this work

[¶]Author for correspondence (melaniegubbelsbupp@rmc.edu)

M.R.G., 0000-0001-9024-3601

Biology Open

across many of these studies is the lack of distinction between naïve and memory T cells.

Studies utilizing mice as models of both acute and chronic malnutrition have shown that T-cell populations decline during malnutrition. Fasting for 48 h reduced total peripheral T-cell numbers in mice, and mice on a 2-week protein-energy malnutrition diet exhibited lower recirculating T-lymphocyte and CD4+ and CD8+ lymphocyte counts than well-nourished mice (Saucillo et al., 2014; Woodward and Miller, 1991). We have previously reported that naïve CD4+ and CD8+ T cells decreased significantly in the lymph nodes and spleen in mice subjected to short-term malnutrition (Murphy et al., 2016). Collins et al. (2019) found a dramatic decrease in the number of memory T cells present in the spleen and lymph nodes over the course of 6 weeks of dietary restriction in a mouse model. In another study of MAL mice, the T-cell population decreased during malnutrition, but the relative abundance of T cells did not decline (Sukhina et al., 2025). Thus, in mice and in the absence of concurrent infections, malnutrition reduces the number of memory and naïve T cells in peripheral lymphoid organs, though not as dramatically as other immune cells.

T-cell function is consistently reduced in malnutrition (Collins et al., 2019; Gerriets and MacIver, 2014; Iyer et al., 2012; Rodríguez et al., 2005; Saucillo et al., 2014; Woodward and Miller, 1991). MAL children with infections had a lower percentage of interleukin (IL)-2- and interferon (IFN)-γ-producing CD4+ T cells than their well-nourished counterparts, while secretion of IL-4 and IL-10 was elevated, and reductions in IL-2 and IFN-γ secretion have been observed in fasted mice (Rodríguez et al., 2005; Saucillo et al., 2014). Thus, T cells have reduced pro-inflammatory cytokine production capabilities during malnutrition, possibly favoring a suite of Th2-associated cytokines (Gerriets and MacIver, 2014). In addition to diminished cytokine production, T cells in MAL mice experience reduced expansion and hastened contraction after infection, indicating an impaired immune response (Sukhina et al., 2025). Memory T cells in MAL mice have also demonstrated a reduced proliferative capacity (Collins et al., 2019; Iyer et al., 2012). Additionally, the migration patterns of MAL T cells may be abnormal; memory T cells were found in higher proportions in the bone marrow during dietary restriction (Collins et al., 2019). Collins et al. (2019) proposed that time in the bone marrow may be key in preserving memory T cells during dietary restriction. It is not known if naïve T cells also experience altered migration during malnutrition.

We have previously reported that levels of corticosterone, the primary glucocorticoid (GC) in mice, are elevated during malnutrition (Murphy et al., 2016). GCs are secreted by the adrenal glands in accordance with a circadian rhythm; levels increase during waking hours and fall during sleeping hours (Zamanian et al., 2013). Stress is also a positive regulator of GCs (Windle et al., 2001). GCs diffuse into cells and interact with GC receptors. Once activated, the receptor moves into the nucleus and interacts with GC-responsive elements, causing both activation and repression of gene transcription (Beato, 1991; Zhang et al., 1997). Transcriptional changes have varying effects on the immune system, ranging from GC-induced immune cell death (Mitchell et al., 1998) to cytokine inhibition (Almawi et al., 1996; Fushimi et al., 1998; Kunicka et al., 1993; Rolfe et al., 1992). In well-nourished organisms, GCs regulate thymocyte development (Taves and Ashwell, 2021), promote the extravasation of naïve T cells from the blood into tissues via upregulation of CXCR4 (Besedovsky et al., 2014), limit inflammatory effector T-cell responses (Brewer et al., 2003; Tehseen et al., 2024), and promote memory T-cell differentiation (Tehseen et al., 2024). During dietary restriction in mice, GCs promote the accumulation of memory T cells in the bone

marrow (Collins et al., 2019). The consequence of malnutrition-induced increases in corticosterone levels on naïve T cells has not been explored.

Most studies examining the relationship between nutrition and the immune system have focused on the activation of the immune system in response to vaccinations or infections (Doeschl-Wilson et al., 2009). The energy required to maintain the immune system in the steady state is difficult to calculate and is certainly orders of magnitude lower than the energetic costs of activating it (Doeschl-Wilson et al., 2009). However, the functionality of an adaptive immune response is first dependent upon the existence of at least a small number of T cells bearing T-cell receptors capable of binding peptides [displayed on major histocompatibility complex (MHC) molecules] derived from the infecting pathogen. Whether and how naïve T-cell diversity is preserved during malnutrition has been understudied. Therefore, this study investigates the effect of malnutrition on the number and location of naïve T cells, the role of GCs in malnutrition-induced alterations to naïve T-cell behavior, and the T-cell-intrinsic and T-cell-extrinsic mechanisms of GC-induced migration and signaling changes. Our results reveal that naïve CD4+ T-cell migration to the bone marrow during malnutrition is intrinsic, requires T-cell sensitivity to GCs, and likely serves to preserve the naïve T-cell population.

## RESULTS

### MAL mice demonstrate decreased numbers of naïve T cells in lymph nodes but increased numbers in the bone marrow

We examined the impact of malnutrition on the size of the naïve T-cell pool in the lymph nodes and bone marrow. Lymph nodes isolated from MAL mice demonstrated a fourfold reduction in the number of naïve CD4+ and CD8+ T cells compared to *ad libitum*-fed (AL) individuals (Fig. 1A,B). However, MAL bone marrow presented with a fourfold to fivefold increase in naïve CD4+ and CD8+ T cells when compared to those from AL individuals (Fig. 1C,D). We additionally confirmed that MAL bone marrow supports increased erythropoiesis (Collins et al., 2019) (Fig. S1A). Given the increased erythropoiesis, we conducted an additional experiment to ensure our red blood cell lysis procedure did not result in differential red blood cell lysis or white blood cell recovery after lysis. We observed little to no red blood cell contamination in either the MAL or control samples after lysis. Further, flow cytometry of MAL and control bone marrow samples that were either subjected to red blood cell lysis or not revealed equivalent cell percentages in the 'live' cell gate, indicating equivalent recovery of nucleated cells in both groups (Fig. S1B). Thus, the increased number of naïve T cells in the bone marrow of MAL mice is unlikely due to differential red blood cell lysis or white blood cell recovery after lysis. Instead, the increased number of naïve T cells in the bone marrow could be due to increased survival of cells in the bone marrow, increased migration to the bone marrow, or both.

### Naïve CD4+ T cells undergo less apoptosis in the bone marrow of MAL mice

The apoptotic rate of naïve T cells in the bone marrow of AL and MAL mice was examined using two different approaches: assessing the presence of Annexin V, coupled with the absence of SYTOX Green to exclude dead cells, and examining the mean fluorescence intensity of intracellular levels of the anti-apoptotic protein Bcl2. When compared to AL bone marrow, MAL bone marrow demonstrated a 2.5-fold decrease in the percentage of Annexin V+ SYTOX Green− naïve CD4+ T cells (Fig. 2A). However, naïve CD8+ T cells in the bone marrow did not display this reduction in apoptosis (Fig. 2B).

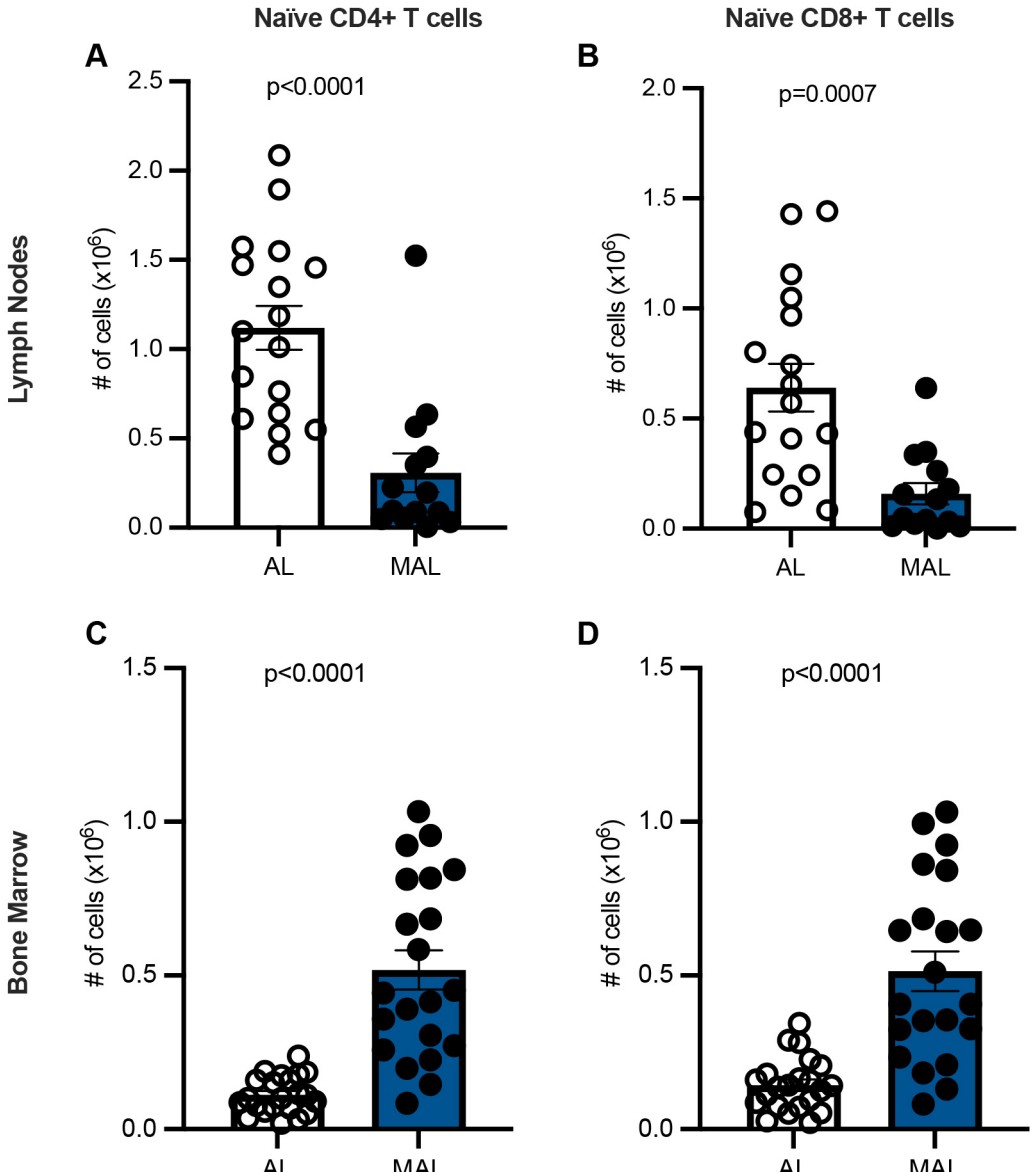

**Fig. 1. During malnutrition, the number of naïve T cells decreases in the lymph nodes but increases in the bone marrow.** (A-D) Flow cytometry was used to determine the percentage of naïve (CD44$^{lo}$) CD4+ (A) and CD8+ (B) T cells in the lymph nodes and naïve (CD44$^{lo}$) CD4+ (C) and CD8+ (D) T cells in the bone marrow. Percentages were multiplied by total numbers to yield the total number of naïve CD4+ and CD8+ T cells. Lymph node: MAL *n*=14, AL *n*=17. Bone marrow: MAL *n*=21, AL *n*=22. *P*-values, obtained by two-tailed unpaired *t*-tests, are indicated above each graph. Graphs display means±s.e.m.

Similarly, the mean fluorescence intensity of Bcl2 displayed a 1.4-fold increase in naïve CD4+ T cells from the bone marrow of MAL mice compared to control cells (Fig. 2C), while there was no difference in the mean fluorescence intensity of Bcl2 between naïve CD8+ T cells from MAL and AL mice (Fig. 2D). MAL memory CD4+ and CD8+ T cells in the bone marrow experienced similar levels of apoptosis compared to control cells (Fig. S2).

**MAL naïve CD4+ T cells preferentially migrate to the bone marrow in an intrinsic manner**

We next examined whether naïve T cells preferentially migrate to the bone marrow at the expense of the lymph nodes during malnutrition. We designed our study to additionally distinguish whether malnutrition intrinsically affects naïve T-cell migration to the bone marrow or rather if the effects we have observed are mediated by non-T cells. We isolated naïve T cells from MAL (Thy1.1−)

and AL (Thy1.1+) mice, mixed them together, labeled them with carboxyfluorescein succinimidyl ester (CFSE), and injected the labeled mixture into AL or MAL recipients. After 2 h, we recovered CFSE+ donor cells from the lymph nodes and bone marrow and determined the percentages that were AL (Thy1.1+) or MAL (Thy1.1−) (Fig. 3A). Finally, we calculated the percent change in the AL:MAL ratio of each cell type between the original, injected cell mixture and the recovered cell mixture (Fig. 3B). MAL naïve CD4+ T cells were less efficient at entering the lymph nodes than control T cells, regardless of the nutrition status of the recipient, as the percent change in the AL:MAL ratio was significantly greater than zero in both AL and MAL recipients (Fig. 3C). MAL naïve CD8+ T cells, on the other hand, were less efficient than AL cells at entering MAL, but not AL, lymph nodes (Fig. 3D). This trend is reversed in the bone marrow, where the percent change in the AL:MAL ratio was significantly less than zero; MAL naïve CD4+ T cells entered the bone marrow more

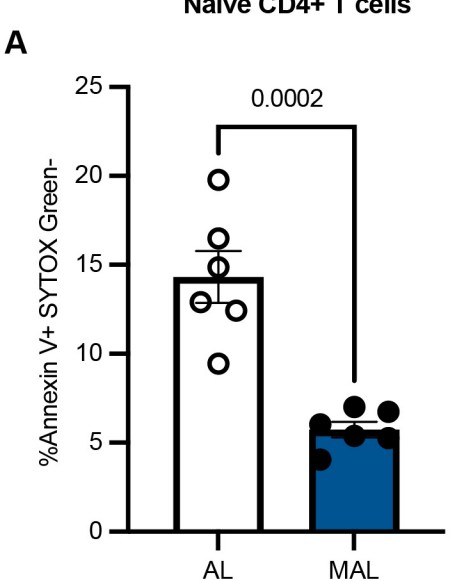

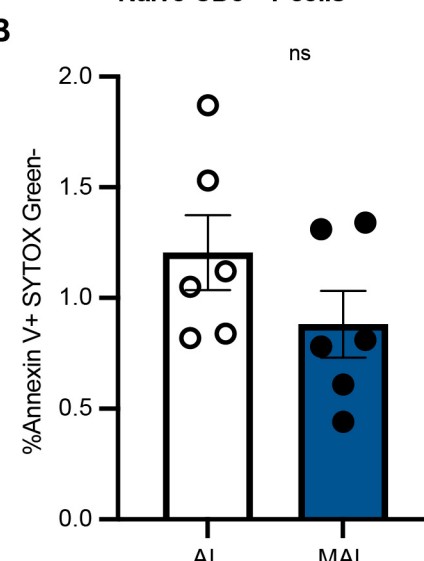

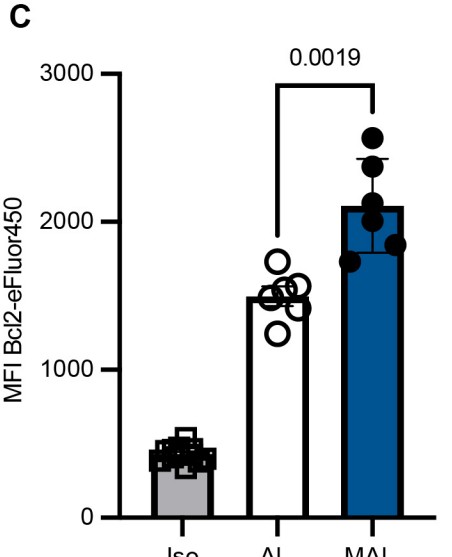

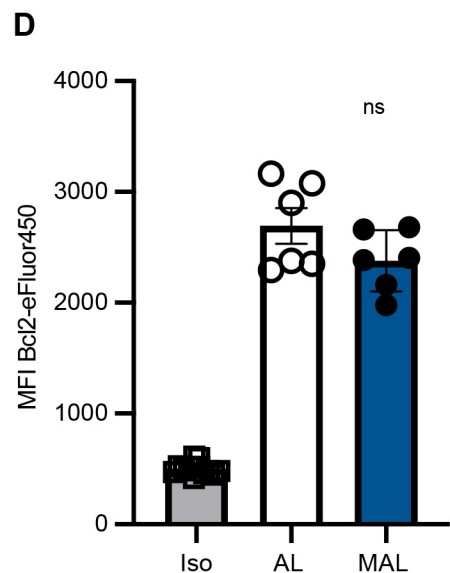

**Fig. 2. Malnourished naïve CD4+ T cells undergo less apoptosis in the bone marrow than control cells.** (A-D) Flow cytometry was used to determine the percentage of Annexin V+ SYTOX Green– naïve (CD44$^{lo}$) CD4+ (A) and CD8+ (B) T cells and the mean fluorescence intensity of Bcl2-eFluor450 of intracellularly stained naïve (CD44$^{lo}$) CD4+ (C) and CD8+ (D) T cells isolated from the bone marrow of MAL and AL mice. Iso, isotype control; MFI, mean fluorescence intensity. $n$=6 per group. $P$-values were obtained by two-tailed unpaired $t$-tests and are indicated above each graph, as applicable. ns, not significant. Graphs display means±s.e.m.

efficiently than AL cells, regardless of the recipient (Fig. 3E). However, the bone marrow of MAL recipients demonstrated a less dramatic percent change in the AL:MAL ratio of naïve CD4+ T cells as compared to AL recipients, perhaps indicating a generally more permissive environment for naïve CD4+ T-cell entry during malnutrition (Fig. 3E). Conversely, MAL naïve CD8+ T cells showed no significant difference from control T cells in their ability to enter bone marrow regardless of the malnutrition status of the recipient; the percent change in the AL:MAL ratio was not significantly different from zero (Fig. 3F). Together, these data indicate that malnutrition-induced redirection of naïve CD4+ T-cell migration to the bone marrow is T-cell intrinsic, with a small, additional impact from T-cell-extrinsic factors.

**T-cell sensitivity to GCs is required and sufficient for naïve CD4+, but not CD8+, T-cell migration to the bone marrow during malnutrition**
We have previously reported significant increases in GC levels during malnutrition (Murphy et al., 2016), and GC levels

impact naïve T-cell migration (Besedovsky et al., 2014). This prompted us to consider the effect of GCs on naïve T-cell migration and chemokine receptor expression during malnutrition, utilizing mice specifically lacking GC receptor expression in T cells [conditional GC receptor knockout (cGRKO)]. In wild-type controls, malnutrition produced the expected increase in the number of naïve CD4+ and CD8+ T cells in the bone marrow (Fig. 4A,B). However, in cGRKO mice, the malnutrition-related increase in bone marrow naïve T-cell numbers was present but greatly diminished relative to AL cGRKO mice (Fig. 4A,B). We also noted that the numbers of CD4+ and CD8+ central and effector memory T cells, as well as CD8+, but not CD4+, resident memory T cells, significantly increased in the bone marrow of wild-type control mice during malnutrition (Fig. S3A-F). As with naïve T cells, the malnutrition-induced increase in the number of effector CD4+ and CD8+ memory as well as central CD4+ memory T cells in the bone marrow was significantly lessened in cGRKO mice, though GC receptor absence had no effect on resident or central CD8+ memory T-cell numbers (Fig. S3A-F).

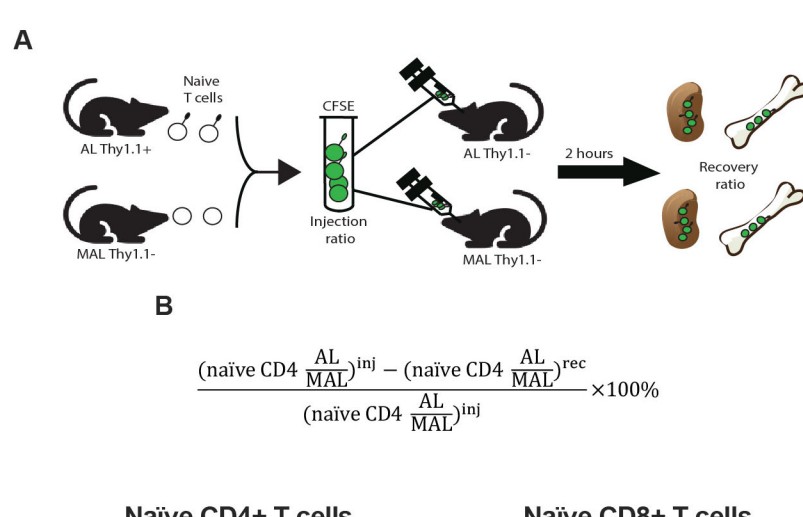

**Fig. 3. Malnutrition intrinsically alters naïve CD4+ T cells to home to the bone marrow.** (A) Pan naïve T cells were isolated from MAL (Thy1.1−) and AL (Thy1.1+) donor mice, mixed in a 50:50 ratio, and labeled with CFSE before retro-orbital injection into AL (Thy1.1−) or MAL (Thy1.1−) recipients. Two hours later, flow cytometry was used to determine the percentage of AL and MAL donor cells in the lymph nodes and bone marrow of recipients. (B) The percent change in the AL:MAL ratio of donor cells in each lymphoid organ was calculated using the equation shown. (C-F) The percent change in the AL:MAL ratio of naïve CD4+ and CD8+ T cells is shown for the lymph nodes (C,D) as well as for the bone marrow (E,F). Lymph node, *n*=10 per group. Bone marrow, *n*=16 per group. Lymph node samples in which the total number of recovered CFSE+ cells was less than 1500 were excluded from further analysis. *P*-values obtained by a one-sample *t*-test, in which the percent change in the AL:MAL ratio is compared to zero, are indicated above each graph, while those obtained by two-tailed unpaired *t*-tests are indicated below each graph. ns, not significant. Graphs display means±s.e.m.

We next examined a possible role for GCs on expression levels of CXCR4 and CCR7, chemokine receptors known to be involved in bone marrow and lymph node homing, respectively (Arieta Kuksin et al., 2015; Yan et al., 2019). CXCR4 was upregulated in MAL naïve CD4+ T cells isolated from the spleen when compared to AL control mice, but malnutrition had no effect on CXCR4 expression in cells isolated from cGRKO mice (Fig. 4C). Malnutrition did not significantly affect CXCR4 expression in naïve CD8+ T cells (Fig. 4D). A similar pattern was observed for CCR7 expression. CCR7 was upregulated in naïve CD4+ T cells residing in the bone marrow of MAL control mice when compared to AL control mice, but a T-cell-specific lack of the GC receptor eliminated the increase (Fig. 4E). Once again, CD8+ naïve T cells showed no significant change in CCR7 expression during malnutrition (Fig. 4F). This indicates that the presence of the GC receptor is critical for malnutrition-induced increases in naïve T-cell migration to the bone marrow. Furthermore, sensitivity to GCs is critical for malnutrition-induced expression changes of

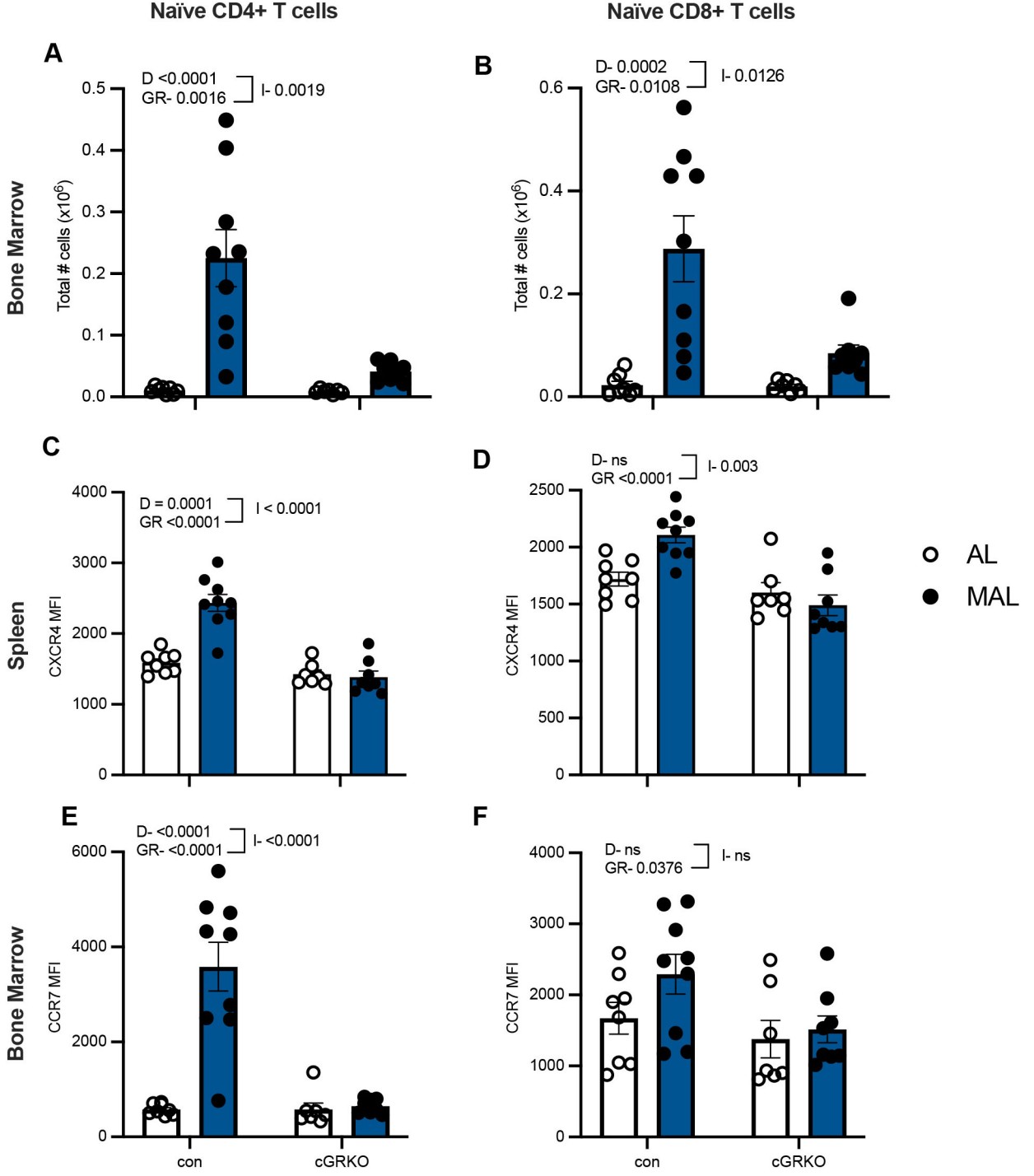

**Fig. 4. T-cell sensitivity to glucocorticoids contributes to malnutrition-induced migration of naïve T cells to the bone marrow.** Cells were isolated from the bone marrow of mice conditionally lacking the glucocorticoid receptor in T cells (cGRKO) or controls (con), and the percentage of naïve (CD44lo CD62Lhi) T cells was identified using flow cytometry. (A,B) Percentages were multiplied by total numbers to yield the number of naïve CD4+ T cells (A) and naïve CD8+ T cells (B). (C-F) In addition, the mean fluorescence intensity of CXCR4 in the spleen (C,D) and CCR7 in the bone marrow (E,F) was measured in naïve CD4+ (C,E) and CD8+ (D,F) T cells by flow cytometry. Control: MAL *n*=9, AL *n*=8. cGRKO: MAL *n*=8, AL *n*=7. *P*-values obtained by two-way ANOVAs are displayed on graphs, with GR indicating the effect of glucocorticoid receptor deficiency, D indicating the effect of diet, and I indicating an interaction between both variables. ns, not significant. Graphs display means±s.e.m.

CCR7 and CXCR4 in CD4+ naïve T cells, despite both AL and MAL naïve CD8+ T cells demonstrating uniform expression. The specific mechanisms facilitating the migration of naïve CD8+ T cells to the bone marrow during malnutrition remain to be determined.

We next injected mice daily with a synthetic corticosteroid, dexamethasone, or control to determine if GCs are sufficient to induce naïve T-cell migration to the bone marrow and affect naïve T-cell chemokine receptor expression. Mice treated with dexamethasone demonstrated twice the number of naïve CD4+

T cells in the bone marrow as controls, although the numbers of naïve CD8+ T cells residing in the bone marrow were similar between dexamethasone-treated and control mice (Fig. 5A,B). Furthermore, naïve CD4+ and CD8+ T cells residing in the spleen both experienced an upregulation of CXCR4 with administration of dexamethasone (Fig. 5C,D). Naïve CD4+ T cells residing in the bone marrow experienced upregulation of CCR7 with dexamethasone treatment (Fig. 5E). On the other hand, naïve CD8+ T cells experienced no change in CCR7 expression when treated with dexamethasone (Fig. 5F). Thus, GCs like dexamethasone are sufficient to change migration behaviors in naïve CD4+ T cells, but not naïve CD8+ T cells. Similarly, dexamethasone treatment was sufficient to upregulate CXCR4 and CCR7 on naïve CD4+ T cells as well as CXCR4 on naïve CD8+ T cells, but not CCR7 on naïve CD8+ T cells. We also assessed whether dexamethasone was sufficient to expand memory T-cell populations in the bone marrow. Indeed, dexamethasone treatment resulted in increased numbers of CD4+ and CD8+ effector memory T cells as well as CD4+ central memory T cells in the bone marrow (Fig. S4). Dexamethasone also enhanced the expression of CXCR4 and CCR7 on several memory T-cell subsets, including CD4+ and CD8+ effector memory T cells (Fig. S4).

## DISCUSSION

In summary, the present study reveals the divergent mechanisms by which malnutrition-induced, GC-dependent signaling reshapes naïve T-cell dynamics. Our analysis revealed a marked redistribution of both naïve CD4+ and CD8+ T cells from the peripheral lymph nodes to the bone marrow. While both subsets require GC sensitivity for migration to the bone marrow, the underlying processes are fundamentally distinct. Naïve CD4+ T cells undergo a mostly cell-intrinsic migratory program, driven by GC-induced upregulation of the chemokine receptors CCR7 and CXCR4, that likely facilitates migration between the spleen and bone marrow. Once in the bone marrow, naïve CD4+ T cells exhibit attenuated rates of apoptosis, suggesting that the bone marrow serves as a protective, survival-promoting niche. In contrast, the accumulation of naïve CD8+ T cells in the bone marrow during malnutrition is not clearly attributable to altered migratory kinetics, increased survival, or chemokine receptor modulation, and notably, GC signaling alone is insufficient to recapitulate these shifts in residency. Collectively, these data demonstrate that malnutrition induces GC-driven chemokine receptor modulation to redirect naïve CD4+ T cells to a protective bone marrow environment.

Our results indicate that increases in the naïve CD4+ T-cell population in the bone marrow during malnutrition occur in a primarily T-cell-intrinsic manner and are directly mediated by GCs. Exposure to natural or synthetic GCs in multiple mammalian models has long been associated with the decline of T cells in the periphery and an accumulation of T cells in the bone marrow (Di Rosa, 2009; Fauci, 1975). More recently, Collins et al. (2019) demonstrated that dietary restriction, which also involves elevated GCs, induces memory T-cell migration to the bone marrow. However, unlike naïve T cells in our model of malnutrition, memory T-cell migration to the bone marrow during dietary restriction did not require T-cell sensitivity to GCs (Collins et al., 2019). Instead, GCs are indirectly involved in memory T-cell migration to the bone marrow during dietary restriction, perhaps playing a role in bone marrow remodeling, which appears to extend the stay of memory T cells (Collins et al., 2019). Though bone marrow remodeling may also impact naïve T-cell migration or residence times in our model, our results indicate that naïve CD4+ T-cell migration to the

bone marrow is primarily due to T-cell-intrinsic changes. Mechanistically, increased expression of CXCR4 on naïve CD4+ T cells in the lymph nodes of MAL mice likely sends the cells to the bone marrow, and increased CCR7 expression on the same cells in the bone marrow directs them back to the lymph nodes or spleen. In well-nourished mice, mTORC2-mediated suppression of CXCR4 is critical for preventing naïve T cells from migrating to the bone marrow (Arojo et al., 2018). Perhaps malnutrition releases naïve CD4+ T cells from this brake on bone marrow homing. Additionally, GCs can enhance T-cell responsiveness to the CXCR4 ligand, CXCL12, in a manner that does not involve upregulation of CXCR4 nor the transcription-modifying activity of GR (Ghosh et al., 2009). The highly elevated levels of serum corticosterone we and others have reported during malnutrition or dietary restriction (Murphy et al., 2016; Collins et al., 2019) may enhance naïve T-cell responsiveness to CXCL12 and result in their migration to the bone marrow. This may partially explain the increase in naïve CD8+ T cells in the bone marrow during malnutrition, even though these cells do not appear to upregulate CXCR4.

The bone marrow may serve as a protective sanctuary for naïve CD4+ T cells during malnutrition. We and others have shown that though the total number of peripheral naïve T cells is diminished during malnutrition, the percentage of T cells in the spleen increased, indicating that T cells are preferentially preserved, at least to some extent, during malnutrition (Murphy et al., 2016; Sukhina et al., 2025). We show here that naïve CD4+ T cells undergo less apoptosis in the bone marrow of MAL mice compared to well-nourished mice, suggesting that the bone marrow environment may shield these cells from cell death. Further, the bone marrow has been shown to be protective for memory T cells during dietary restriction: the migration of CD8+ central memory T cells to the bone marrow during dietary restriction was associated with reduced apoptosis and improved memory cell function (Collins et al., 2019). We did not observe this phenomenon; however, we used only CD44[hi] to distinguish memory T cells rather than subdividing the memory T-cell population into different subsets. Additionally, Collins et al. (2019) exclusively utilized female mice, while our studies used mostly male mice; this is worth considering as sex plays a strong role in immunity (Klein and Flanagan, 2016). If an adaptive mechanism is at play in preserving naïve T cells during malnutrition, one would expect that refeeding would restore T-cell numbers and function. Indeed, refeeding mice that were previously MAL resulted in a full recovery of their T-cell populations, including normal T-cell expansion and cytokine production during infection (Sukhina et al., 2025). This protection may be reserved for T cells alone, as refeeding did not completely restore the ability to expand neutrophil populations in the context of infection (Sukhina et al., 2025).

The dramatic increase in the total number of naïve T cells in the bone marrow of MAL mice is striking and prompts consideration of what adaptive benefits may result. One possibility is that T-cell presence in the bone marrow influences hematopoiesis to prioritize the production of specific cell types needed during caloric and nutrient deficiencies. Supportively, activated T cells coordinate peripheral demand for particular immune cells with their development in the bone marrow during infection (Bonomo et al., 2016). In the absence of infection, T-cell-deficient mice exhibit defective myelopoiesis and erythropoiesis, and the restoration of CD4+ T cells rescues myelopoiesis, but not erythropoiesis. Myelopoiesis was only restored when the CD4+ T cells were activated in an antigen-specific manner (Monteiro et al., 2005). We and others noted the dramatic increase in erythropoiesis during malnutrition or dietary restriction along with increases in the number of T cells in the bone

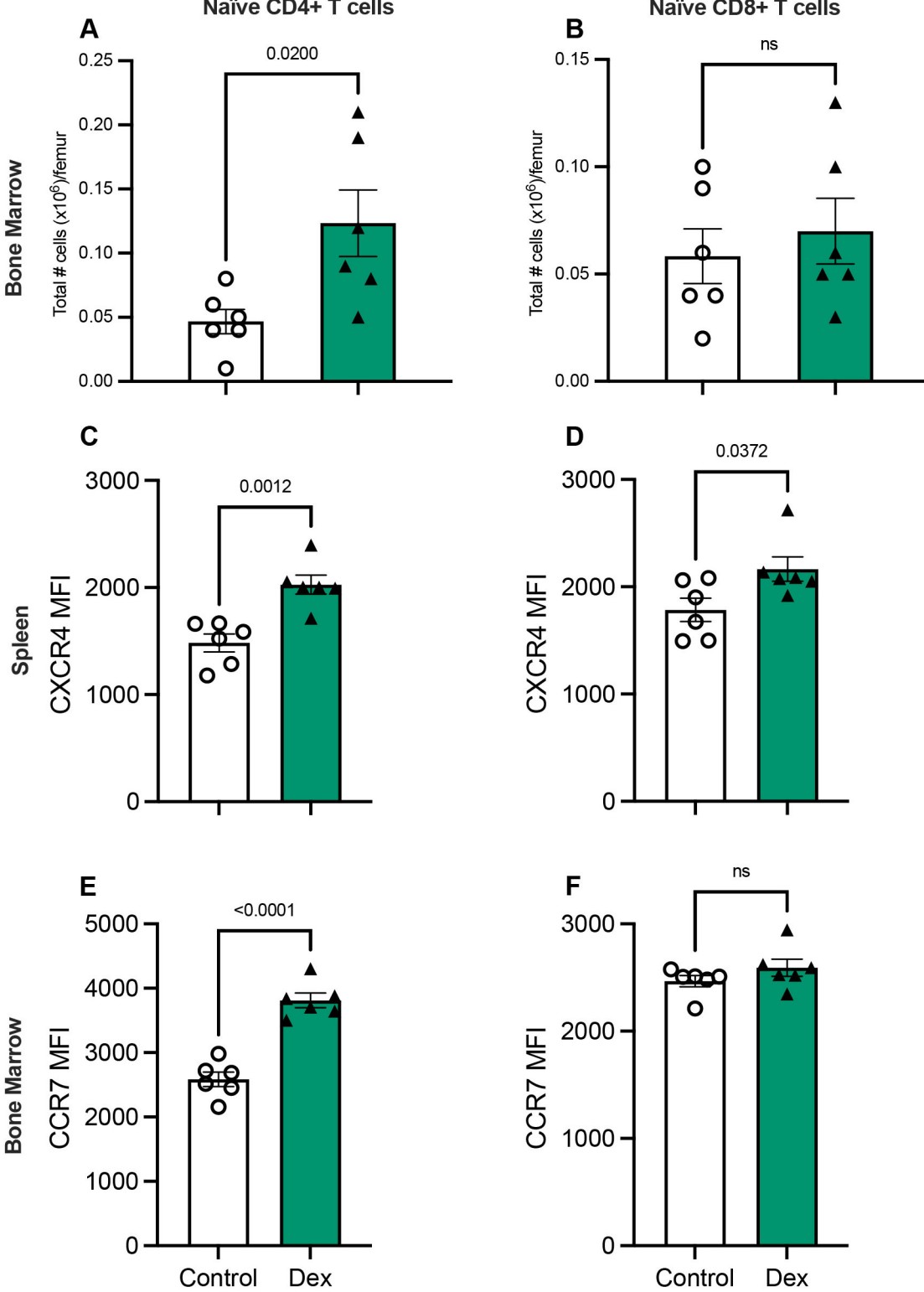

**Fig. 5. Glucocorticoids are sufficient to induce naïve CD4+ T-cell migration to the bone marrow.** Cells were isolated from the bone marrow of mice injected with dexamethasone (2 mg kg$^{-1}$, resuspended in DMSO and diluted with sterile saline) or control. (A,B) Percentages of naïve (CD44$^{lo}$ CD62L$^{hi}$) T cells were identified using flow cytometry and multiplied by total numbers to yield the number of naïve CD4+ T cells (A) and naïve CD8+ T cells (B). (C-F) In addition, the mean fluorescence intensity of CXCR4 in naïve CD4+ T cells (C) and naïve CD8+ T cells (D) from the spleen and CCR7 in naïve CD4+ T cells (E) and naïve CD8+ T cells (F) from the bone marrow was measured by flow cytometry. $n$=6 per group. Individual values are represented by black triangles (dexamethasone) or open circles (control). $P$-values were obtained by two-tailed unpaired $t$-tests and are displayed above the corresponding graph. ns, not significant. Graphs display means±s.e.m.

marrow (Collins et al., 2019). In both studies, the number of T cells in the bone marrow was elevated. Together, these observations support the hypothesis that T-cell presence in the bone marrow enhances erythropoiesis but does not reveal how or why such a mechanism may have evolved. During food deprivation, organisms exhibit increased locomotor activity, which is likely an adaptation that enables foraging behavior (Dietrich et al., 2015). Perhaps T-cell-mediated enhanced erythropoiesis during undernutrition enables greater physical activity, even in the face of reduced nutrition and caloric intake, for more effective foraging.

A study involving the modeling of evolutionary incentives for energy invested in immunity showed that investment hinges on the availability of food. When food is not consistently available, it is more favorable to reduce the allocation of energy to immune cells and instead stockpile energy reserves to promote foraging (Houston et al., 2007). Minimizing the energy consumption of naïve T cells could increase the availability of energy for investment in finding food during times of scarcity. We hypothesize that MAL naïve T cells are 'super quiescent'. Naïve T cells are normally quiescent, expending little energy prior to activation (Hamilton and Jameson, 2012). However, the impairment of MAL T cells' glucose uptake, migration speed, expansion, and cytokine secretion after activation indicates that MAL T cells may be in an abnormal state (Collins et al., 2019; Iyer et al., 2012; Saucillo et al., 2014; Sukhina et al., 2025). This state of 'super quiescence', or ultra-low-energy consumption leading to sluggishness after activation, could be a protective mechanism allowing for preservation during extreme nutrient shortages seen in malnutrition, trading function for preservation. During super quiescence, naïve T cells may exhibit reduced energy utilization while enjoying support from bone marrow cells and protection from apoptosis. In addition to energy conservation, the maintenance of adequate numbers of naïve T cells with a sufficient variety of T-cell receptors in the bone marrow during malnutrition for use after malnutrition ends could have contributed to the selection of organisms with super quiescent naïve T cells in the bone marrow during nutritional stress.

Overall, we have described the effects of malnutrition on naïve T cells and implicated GCs in mediating these changes. This mechanism may protect the future immune response and induce a state of super quiescence, reducing the energy cost of naïve T-cell surveillance and preserving energy for foraging behaviors necessary for survival. The malnutrition-related increase in naïve T-cell residency of the bone marrow likely preserves naïve T-cell numbers until nutrition is restored, though further studies examining the impact of bone marrow residency on naïve T-cell function will be informative.

## MATERIALS AND METHODS
### Mice (*Mus musculus*)
Mice were singly housed in Randolph-Macon College's mouse facility. C57BL/6J and B6 Thy1.1 mice were obtained from the Jackson Laboratory (Maine, USA). Nr3c1$^{fl/fl}$ and CD4.cre+ mice were purchased from The Jackson Laboratory (Maine, USA) and crossed in-house to obtain Nr3c1$^{fl/fl}$CD4.cre+ and Nr3c1$^{fl/fl}$CD4.cre− mice. Nr3c1$^{fl/fl}$ CD4.cre+ mice, hereafter referred to as cGRKO mice, lack GC receptors in T cells. Treatment of animals complied with the standards set by the Institutional Animal Care and Use Committee of Randolph-Macon College (protocol no. 03-2025) and the National Institutes of Health's Guide for the Care and Use of Laboratory Animals. Mice were randomly assigned to either an AL control diet or a MAL diet. Chow (Teklad Global 18% Protein Rodent Diet, Harlan Laboratories) consumption was tracked for a period of 2 weeks. AL mice were permitted unlimited access to chow (100 g per week), while MAL mice were fed the lesser of two options: 35% less chow by weight than eaten over the previous 2-week period or 13 g of chow. After 1 week of the malnutrition diet, mice were euthanized at 10 weeks of age using $CO_2$ overdose and subsequent

cervical dislocation. On average, chow consumption decreased by 50% for MAL mice. MAL mice lost an average of about 15% of their initial body weight while AL mice gained about 3.9%. The malnutrition protocol is akin to dietary restriction models that restrict food access by 50% but is distinct from calorie restriction (Asami et al., 2022; Collins et al., 2019). Only male mice were included in studies, except for the adoptive transfer and cGRKO experiments, in which almost equal numbers of males and females were included. The male:female ratio for the adoptive transfer study was 1.08, and for the cGRKO study, it was 0.9. For the dexamethasone study, C57BL/6J mice received daily intraperitoneal injections of dexamethasone (2 mg kg$^{−1}$, resuspended in DMSO and diluted with sterile saline) or a similar DMSO/ saline mixture for 1 week. Dexamethasone-injected mice lost an average of about 5.4% of their initial body weight, while mice injected with DMSO mixed with saline lost about 0.51%.

### Flow cytometry
After euthanasia, cells were isolated from mouse femoral bone marrow and axillary, brachial, and inguinal lymph nodes; lymph node cells were pooled following isolation. Ammonium-chloride-potassium (ACK) lysis buffer (0.15 M $NH_4Cl$; 10 mM $KHCO_3$; 0.1 mM EDTA) was used to remove red blood cells from the bone marrow samples. Cells were counted with a hemocytometer, and Trypan Blue was utilized to distinguish live cells from dead. After cells were incubated with an Fc-shielding reagent (Purified Anti-Mouse CD16/Cd32, clone 2.4G2, 1:5 dilution, cat no. 70-0161, Cytek Biosciences), the following markers were used to differentiate T cells: naïve [CD62L (clone MEL-4, 1:100 dilution, cat no. 104440, BioLegend) and/or CD44 (clone IM7, 1:400 dilution, cat no. 80-0441, Cytek Biosciences)], CD4+ T cell [CD4 (clone GK1.5, 1:200 dilution, cat no. 25-0041, Thermo Fisher)], CD8+ T cell [CD8 (clone 53-6.7, 1:100 dilution, cat no. 560471, Waters Biosciences)]. Annexin V-Pacific Blue (cat no. A35122, Thermo Fisher), SYTOX Green (cat no. R37168, Thermo Fisher), and antibodies to Thy1.1 (clone HIS51, 1:200 dilution, cat no. A16413, Invitrogen) and CXCR4 (clone 2B11, 1:200 dilution, cat no. 53-9991, eBioscience), CCR7 (clone 4B12, 1:200 dilution, cat no. 12-1971, eBioscience), and Bcl2 (clone 10C4, 5 μl per test, cat no. 48-6992-42, eBioscience) were also used to stain cells. Naïve T cells were distinguished by CD44$^{lo}$ in all experiments. Additionally, high expression of CD62L was used in studies with cGRKO and dexamethasone-treated mice. Cells were washed after staining and subsequently acquired on a NovoCyte 3005 flow cytometer (Agilent Technologies, California, USA). NovoExpress software was used for data analysis.

### Adoptive transfer
An EasySep™ Mouse Pan-Naïve T-Cell Isolation Kit (STEMCELL Technologies, Vancouver, British Columbia, Canada) was used to isolate naïve T cells from AL Thy1.1+ and MAL Thy1.1− donor mice. After equal numbers of cells from each donor population were pooled and labeled with 5 mM CFSE (VWR), $10×10^6$ labeled cells were retro-orbitally injected into AL Thy1.1− and MAL Thy1.1− recipient mice. Prior to injection, the ratio of AL:MAL donor cells was calculated for naïve CD4+ and CD8+ T cells by flow cytometry. Recipient mice were euthanized 2 h after injection to allow the T cells time to disperse, and flow cytometry was conducted after isolating cells from the lymph nodes and bone marrow as previously described. The ratio of AL:MAL donor cells recovered in the lymph node and bone marrow was calculated for naïve CD4+ and CD8+ T cells. CFSE distinguished donor cells from recipient cells, and Thy1.1 staining distinguished AL (Thy1.1+) from MAL (Thy1.1−) donor cells. The total number of injected and recovered cells is shown (Fig. S5). To normalize the data from two independent experiments and T-cell subsets, the percent change in the AL:MAL ratio of naïve CD4+ and CD8+ T cells between the original, injected cell mixture and the recovered cell mixture was calculated as follows:

$$\frac{\left(\text{naïve CD4 } \frac{\text{AL}}{\text{MAL}}\right)^{\text{inj}} - \left(\text{naïve CD4 } \frac{\text{AL}}{\text{MAL}}\right)^{\text{rec}}}{\left(\text{naïve CD4 } \frac{\text{AL}}{\text{MAL}}\right)^{\text{inj}}} \times 100\%.$$

## Statistical analysis

Two-tailed unpaired *t*-tests, two-tailed one-sample *t*-tests, and two-way ANOVAs, calculated in GraphPad Prism, were used to determine the statistical significance of differences among treatment groups ($P<0.05$). One-sample *t*-tests were conducted to determine if the percent change in the AL:MAL ratio in adoptive transfer studies was significantly different from zero. Unpaired *t*-tests were conducted to compare experimental groups to each other, while two-way ANOVAs were conducted to determine the statistical significance of two variables in one experiment.

## Acknowledgements

We thank Charles Gowan for his expert statistical advice. We also thank Syreen Goulmamine, David Gibson, Aja Washington, Marissa Marczak, Rithanya Saravanan, and Caroline Pearson for their technical assistance. Finally, we thank the Randolph-Macon College mouse care workers for their expert care of research mice.

## Competing interests

The authors declare no competing or financial interests.

## Author contributions

Conceptualization: M.R.G.B.; Data curation: M.R.G.B.; Formal analysis: M.L.S., J.H., M.R.G.B.; Funding acquisition: M.R.G.B.; Investigation: M.L.S., J.H., T.R.F., C.P., K.A.D., M.R.G.B.; Methodology: M.R.G.B.; Resources: M.R.G.B.; Supervision: M.R.G.B.; Writing – original draft: M.L.S., J.H., M.R.G.B.; Writing – review & editing: M.R.G.B.

## Funding

This work was funded by two National Science Foundation grants to M.R.G.B. – Major Research Instrumentation (1920116) and Research at Undergraduate Institutions (1951881) – and also by a Rashkind grant from Randolph-Macon College. Open Access funding provided by National Science Foundation. Deposited in PMC for immediate release.

## Data and resource availability

All relevant data and details of resources can be found within the article and its supplementary information.

## Peer review history

The peer review history is available online at https://journals.biologists.com/bio/lookup/doi/10.1242/bio.062485.reviewer-comments.pdf

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
