## [Peer Review File · Biology Open]

Glucocorticoids intrinsically redirect naïve CD4+ T cells to the bone marrow for preservation in malnourished mice

Madeline L. Smith, Jacob Hanes, Takesha R. Foster, Clay Phillips, Kwesi A. Dadzie and Melanie R. Gubbels Bupp
10.1242/bio.062485

Editor: Christopher A. Maher

Review timeline

Original submission:	12 January 2026
Editorial decision:	19 January 2026
First revision received:	31 March 2026
Accepted:	6 April 2026

Original submission

First decision letter

MS ID#: bio.062485

MS Title: Glucocorticoids intrinsically redirect naïve CD4+ T cells to the bone marrow for preservation in malnourished mice

Authors: Madeline L. Smith, Jacob Hanes, Takesha R. Foster, Clay Phillips, Kwesi A. Dadzie and Melanie R. Gubbels Bupp

I have now reached a decision on the above manuscript.

The reviewer reports are shown at the bottom of this email.

As you will see, the reviewers raised a number of substantial criticisms that prevent me from accepting the paper at this stage.

They suggest, however, that a revised version might prove acceptable, if you can address their concerns. If you think that you can deal satisfactorily with the criticisms on revision, I would be pleased to see a revised manuscript. We would then return it to the reviewers.

At this stage, we also ask you to ensure your manuscript complies with our formatting guidelines. Provided you are able to fully address the referees' comments, we are positive about publication of your paper (we accept over 95% of revision submissions) and therefore hope you won't mind any extra work involved in reformatting your manuscript at this point.

Please upload both a 'clean' version of your Word file, along with a highlighted version clearly showing where you have made changes in the revised manuscript. Please avoid using 'Track changes' in Word files as these are lost in PDF conversion.

I should be grateful if you would also provide a point-by-point response detailing how you have dealt with the points raised by the reviewers in the 'Response to Reviewers' box. Please attend to all of the reviewers' comments. If you do not agree with any of their criticisms or suggestions please explain clearly why this is so.

Reviewer 1

Comments for the author

Madeline L. Smith and colleagues addresses how malnutrition alters naïve T cell distribution and proposes a T cell-intrinsic, glucocorticoid-dependent redirection of naïve CD4⁺ T cells to bone marrow, linked to CXCR4/CCR7 changes and reduced apoptosis. The paper is well written, and all the controls are in place. The concept is interesting. I have only two minor concerns before publication:

1) Absolute quantification of bone marrow naïve T cells relies on lysis and hemocytometer counts without bead-based normalization. Because nutritional status can alter erythropoiesis/bone marrow cellularity, lysis efficiency and nucleated cell recovery may differ between diet groups. Please address this limitation.

2) The manuscript infers that endogenous glucocorticoids drive CXCR4/CCR7 upregulation primarily from glucocorticoid receptor deletion. While this supports glucocorticoid receptor dependence, it does not directly establish that glucocorticoids are necessary and sufficient for the receptor changes. Testing necessity (e.g., lowering endogenous glucocorticoids) and/or sufficiency (exogenous glucocorticoid administration) would be required to attribute the effect specifically to glucocorticoids rather than other malnutrition-associated cues that may be indirectly blocked by glucocorticoid receptor deletion.

Reviewer 2:

Comments for the author

In their manuscript, the authors explore the impact of malnutrition on naïve T cells, linking glucocorticoid levels and T cell migration to various tissues. The work builds on prior studies that reported elevated glucocorticoid levels in malnourished individuals and mice. Malnutrition increases the level of glucocorticoids, impacting CXCR4 and CCR7 expression in CD4 naïve T cells, promoting their migration to the bone marrow. They also observed an increased migration of CD T cells in these mice, but the mechanisms here are poorly justified. The work is, in general, properly controlled, and the results are supported by relevant data.

Overall, the work is built logically, and most conclusions are supported by data. I have a few suggestions here, which I hope will be helpful to the authors.

Main elements:

1- I would be careful when concluding that the T cells experience less apoptosis in the bone marrow (Figure 2). The authors should use an additional method to monitor apoptosis (e.g., caspase-3 cleavage, cleavage of caspase substrates, Tunel, etc), as recommended by international cell death committee recommendations. In a certain context, Annexin V can be exposed on cells that do not undergo apoptosis. Propidium iodine staining or any life/dead stain would be appropriate here to stain cells and assess cell death as well (in addition to Annexin V). The basal cell death level observed in the bone marrow (20-30% for T cells) seems high to me.

2- In general, the involvement of CXCR4 and CCR7 in CD4 T cells seems to be a consequence of the glucocorticoid level in MAL mice. However, the results in CD8 T cells do not support their conclusions and their migration to the bone marrow during MAL. The authors may either wish to add additional information that explains why CD8 T cells migrate more to the bone marrow or tone down their conclusions. I feel the results sections fail to do this.

Minor elements:

1- This may be an issue on my computer, but the resolution of the graphs and figures presented is not ideal and should be improved.

2- Figure 3: Make sure there is a label on the y axis. It may also be preferable to relabel the panels as they did in Figure 2.

3- In the method section, mention the ratio of male and female mice used for the work.

4- In figure 3, I would mention the number of cells recovered in addition to the ratio. This may be added as a supplemental figure, but I believe it is important to properly evaluate the results obtained in the figure.

5- The combination of statistical tests used should be better justified in the text. Also, in Figure 3, it is unclear where Welch's test has been performed. This is mentioned in the figure legend but not referenced in the figure itself. I am unsure why such combinations of test varies across the manuscript.

6- Justify in Figure 3 why you did not use a paired number of lymph node and bone marrow samples.

Reviewer's Responses to Questions

Experimental quality

Does each figure have the proper controls?

If 'No', please indicate reasons in Comments for Author box below.

Reviewer #1:

- Yes

Reviewer #2:

- Yes

Were the data analyzed using appropriate statistical tests?

If 'No', please indicate reasons in Comments for Author box below.

Reviewer #1:

- Yes

Reviewer #2:

- No

Reproducibility

Were experiments performed using adequate number of biological replicates?

If 'No', please indicate reasons in Comments for Author box below.

Reviewer #1:

- Yes

Reviewer #2:

- Yes

Does the methods section provide sufficient detail to permit reproducibility?

If 'No', please indicate reasons in Comments for Author box below.

Reviewer #1:

- Yes

Reviewer #2:

- Yes

Completeness

Are the manuscript's conclusions supported by the data?

If 'No', please indicate reasons in Comments for Author box below.

Reviewer #1:

- Yes

Reviewer #2:

- No

Scholarship

Do the authors cite and discuss the merits of data that would argue for and against their conclusion?

If 'No', please indicate reasons in Comments for Author box below.

Reviewer #1:

- Yes

Reviewer #2:

- Yes

Does the manuscript title & abstract accurately reflect the contents of the manuscript, without hyperbole?

If 'No', please indicate reasons in Comments for Author box below.

Reviewer #1:

- Yes

Reviewer #2:

- Yes

First revision

Author response to reviewers' comments

We believe we have addressed the reviewers' suggestions and comments in the attached revised manuscript. The manuscript is now stronger, and we thank the reviewers for their thoughtful suggestions.

Comments from Reviewer #1:

1) Absolute quantification of bone marrow naïve T cells relies on lysis and hemocytometer counts without bead-based normalization. Because nutritional status can alter erythropoiesis/bone marrow cellularity, lysis efficiency and nucleated cell recovery may differ between diet groups. Please address this limitation.

We included the following language in the text of the results section on pages 5 and 6, and included a supplementary figure demonstrating equivalent nucleated cell recovery after lysis.

“We additionally confirmed that malnourished bone marrow supports increased erythropoiesis (Collins et al., 2019) (Fig. S1A). Given the increased erythropoiesis, we conducted an additional experiment to ensure our red blood cell lysis procedure did not result in differential red blood cell lysis or white blood cell recovery after lysis. We observed little to no red blood cell contamination in either the malnourished or control samples after lysis. Further, flow cytometry of malnourished and control bone marrow samples that were either subjected to red blood cell lysis or not revealed equivalent cell percentages in the “live” cell gate, indicating equivalent recovery of nucleated cells in both groups (Fig. S1B). Thus, the increased number of naïve T cells in the bone marrow of malnourished mice is unlikely due to differential red blood cell lysis or white blood cell recovery after lysis. Instead, the increased number of naïve T cells in the bone marrow could be due to increased survival of cells in the bone marrow, increased migration to the bone marrow, or both.”

2) The manuscript infers that endogenous glucocorticoids drive CXCR4/CCR7 upregulation primarily from glucocorticoid receptor deletion. While this supports glucocorticoid receptor dependence, it does not directly establish that glucocorticoids are necessary and sufficient for the receptor changes. Testing necessity (e.g., lowering endogenous glucocorticoids) and/or sufficiency (exogenous glucocorticoid administration) would be required to attribute the effect

specifically to glucocorticoids rather than other malnutrition-associated cues that may be indirectly blocked by glucocorticoid receptor deletion.

We appreciate this suggestion and have added a study to the manuscript in which we have treated well-nourished mice with exogenous synthetic glucocorticoid (dexamethasone) or control. The results are shown in Figure 5. The description of the results are on page 8 and are also pasted below.

“We next injected mice daily with a synthetic corticosteroid, dexamethasone, or control to determine if GCs are sufficient to induce naïve T cell migration to the bone marrow and affect naïve T cell chemokine receptor expression. Mice treated with dexamethasone demonstrated twice the number of naïve CD4⁺ T cells in the bone marrow as controls, although the numbers of naïve CD8⁺ T cells residing in the bone marrow was similar between dexamethasone-treated and control mice (Figs 5A,B). Furthermore, naïve CD4⁺ and CD8⁺ T cells residing in the spleen both experienced an upregulation of CXCR4 with administration of dexamethasone (Figs 5C,D). Naïve CD4⁺ T cells residing in the bone marrow experienced upregulation of CCR7 with dexamethasone treatment (Fig. 5E). On the other hand, naïve CD8⁺ T cells experienced no change in CCR7 expression when treated with dexamethasone (Fig. 5F). Thus, GCs like dexamethasone are sufficient to change migration behaviors in naïve CD4⁺ T cells, but not naïve CD8⁺ T cells. Similarly, dexamethasone treatment was sufficient to upregulate CXCR4 and CCR7 on naïve CD4⁺ T cells as well as CXCR4 on naïve CD8⁺ T cells, but not CCR7 on naïve CD8⁺ T cells. We also assessed if dexamethasone was sufficient to expand memory T cell populations in the bone marrow. Indeed, dexamethasone treatment resulted in increased numbers of CD4⁺ and CD8⁺ effector memory T cells as well as CD4⁺ central memory T cells in the bone marrow (Fig. S4). Dexamethasone also enhanced the expression of CXCR4 and CCR7 on several memory T cell subsets, including CD4⁺ and CD8⁺ effector memory T cells (Fig. S4).”

The methods section now contains a sentence describing this study on page 12 and pasted below: “For the dexamethasone study, C57BL/6J mice received daily intraperitoneal injections of dexamethasone (2 mg kg⁻¹), resuspended in DMSO and diluted with sterile saline) or a similar DMSO/saline mixture for one week. Dexamethasone-injected mice lost an average of about 5.4% of their initial body weight while mice injected with DMSO mixed with saline lost about 0.51%.”

Comments from Reviewer #2:

I would be careful when concluding that the T cells experience less apoptosis in the bone marrow (Figure 2). The authors should use an additional method to monitor apoptosis (e.g., caspase-3 cleavage, cleavage of caspase substrates, Tunel, etc), as recommended by international cell death committee recommendations. In a certain context, Annexin V can be exposed on cells that do not undergo apoptosis. Propidium iodine staining or any life/dead stain would be appropriate here to stain cells and assess cell death as well (in addition to Annexin V). The basal cell death level observed in the bone marrow (20-30% for T cells) seems high to me.

We agree and thank the reviewer for this suggestion. We have added two additional studies to strengthen our conclusion regarding apoptosis—first, we stained bone marrow cells with SYTOX Green and Annexin V together. Secondly, we also assessed whether malnourished cells expressed more of the anti-apoptotic protein, Bcl-2, than controls. The results of these studies have replaced the previous Figure 2 and are described in the text of the results on page 6 (also pasted below). Overall, using both methods to assess apoptosis, we have shown that malnourished naïve CD4⁺ T cells in the bone marrow experience less apoptosis than control cells, but naïve CD8⁺ T cells experience approximately similar rates of apoptosis between the two groups.

“The apoptotic rate of naïve T cells in the bone marrow of *ad libitum*-fed (AL) and malnourished (MAL) mice was examined using two different approaches: assessing the presence of Annexin V, coupled with the absence of SYTOX Green to exclude dead cells, as well as the mean fluorescence intensity of intracellular levels of the anti-apoptotic protein, Bcl2. When compared to *ad libitum* bone marrow, malnourished bone marrow demonstrated a 2.5-fold decrease in the percentage of Annexin V+ SYTOX Green- naïve CD4⁺ T cells (Fig. 2A). However, naïve CD8⁺ T cells in the bone marrow did not display this reduction in apoptosis (Fig. 2B). Similarly, the mean fluorescence intensity of Bcl2 displayed a 1.4-fold increase in naïve CD4⁺ T cells from the bone marrow of malnourished mice compared to control cells (Fig. 2C), while there was no difference in the mean

fluorescence intensity of Bcl2 between naïve CD8+ T cells from MAL and AL mice (Fig. 2D). Malnourished memory CD4+ and CD8+ T cells in the bone marrow experienced similar levels of apoptosis compared to control cells (Fig. S2).”

In general, the involvement of CXCR4 and CCR7 in CD4 T cells seems to be a consequence of the glucocorticoid level in MAL mice. However, the results in CD8 T cells do not support their conclusions and their migration to the bone marrow during MAL. The authors may either wish to add additional information that explains why CD8 T cells migrate more to the bone marrow or tone down their conclusions. I feel the results sections fail to do this.

We agree and included the following sentence on page 8, lines 6-8, of the results section:

“The specific mechanisms facilitating the migration of naïve CD8+ T cells to the bone marrow during malnutrition remain to be determined.”

We also revised the first paragraph of the discussion section (page 8-9) to make the distinction between mechanisms for naïve CD4+ and CD8+ T cell migration to the bone marrow clearer. It is also pasted below:

“In summary, the present study reveals the divergent mechanisms by which malnutrition-induced, glucocorticoid-dependent signaling reshapes naïve T cell dynamics. Our analysis revealed a marked redistribution of both naïve CD4+ and CD8+ T cells from the peripheral lymph nodes to the bone marrow. While both subsets require GC sensitivity for migration to the bone marrow, the underlying processes are fundamentally distinct. Naïve CD4+ T cells undergo a mostly cell-intrinsic migratory program, driven by GC-induced upregulation of the chemokine receptors CCR7 and CXCR4, that likely facilitates migration between the spleen and bone marrow. Once in the bone marrow, naïve CD4+ T cells exhibit attenuated rates of apoptosis, suggesting that the bone marrow serves as a protective, survival-promoting niche. In contrast, the accumulation of naïve CD8+ T cells in the bone marrow during malnutrition is not clearly attributable to altered migratory kinetics, increased survival, or chemokine receptor modulation, and notably, GC signaling alone is insufficient to recapitulate these shifts in residency. Collectively, these data demonstrate that malnutrition induces GC-driven chemokine receptor modulation to redirect naïve CD4+ T cells to a protective bone marrow environment.”

Minor elements from reviewer #2:

1- This may be an issue on my computer, but the resolution of the graphs and figures presented is not ideal and should be improved.

We apologize and have submitted high-quality, high-resolution graphs and figures.

2- Figure 3: Make sure there is a label on the y axis. It may also be preferable to relabel the panels as they did in Figure 2.

We have included y axis labels on all panels in the manuscript.

3- In the method section, mention the ratio of male and female mice used for the work.

This was addressed on page 12, lines 8-11 and is also pasted below:

“Only male mice were included in studies, except for the adoptive transfer and cGRKO experiments, in which almost equal numbers of males and females were included. The male:female ratio for the adoptive transfer study was 1.08 and for the cGRKO study it was 0.9.”

4- In figure 3, I would mention the number of cells recovered in addition to the ratio. This may be added as a supplemental figure, but I believe it is important to properly evaluate the results obtained in the figure.

We have included the total number of cells recovered as supplemental figure 5. However, we should point out that the total number of cells is not informative for comparisons, because a pan

naïve T cell isolation kit was utilized to obtain the donor cells. Pan naïve T cell isolation kits capture all naïve T cells in the endogenous CD4:CD8 T ratio of the donors. The ratio of the total number of MAL and AL injected cells was 50:50, but the ratio of AL naïve CD4+ T cells to MAL naïve CD4+ T cells, for example, is not 50:50 and differed between experiments. The percent change in the ratio that is graphed in Figure 3 accommodates the different number of injected naïve CD4+ and naïve CD8+ T cells and also normalizes the data from two independent experiments so that they can be meaningfully combined in one graph.

We also added the following to the methods section on page 13, lines 11-17:

“The total number of injected and recovered cells are shown (Fig S5). To normalize the data from two independent experiments and T cell subsets, the percent change in the AL:MAL ratio naïve CD4+ and CD8+ T cells between the original, injected cell mixture and the recovered cell mixture was calculated using Eqn 1:

$$\frac{(\text{naïve CD4 } \frac{\text{AL}}{\text{MAL}})^{\text{inj}} - (\text{naïve CD4 } \frac{\text{AL}}{\text{MAL}})^{\text{rec}}}{(\text{naïve CD4 } \frac{\text{AL}}{\text{MAL}})^{\text{inj}}} \times 100\%,$$

(1)”

5- The combination of statistical tests used should be better justified in the text. Also, in Figure 3, it is unclear where Welch’s test has been performed. This is mentioned in the figure legend but not referenced in the figure itself. I am unsure why such combinations of test varies across the manuscript.

The Welch’s t tests have been replaced with unpaired t tests throughout the manuscript and the text of the figure legends has been updated appropriately. The following language was also added to the methods section:

“Two-tailed unpaired *t*-tests, two-tailed one-sample *t*-tests, and two-way ANOVAs, calculated in GraphPad Prism, were used to determine the statistical significance of differences among treatment groups ($\alpha < 0.05$). One-sample *t*-tests were conducted to determine if the percent change in the AL:MAL ratio in adoptive transfer studies was significantly different from zero. Unpaired *t*-tests were conducted to compare experimental groups to each other, while two-way ANOVAs were conducted to determine the statistical significance of two variables in one experiment.”

6- Justify in Figure 3 why you did not use a paired number of lymph node and bone marrow samples.

The following sentence was added to the Figure legend 3:

“Lymph node samples in which the total number of recovered CFSE+ cells was less than 1,500 were excluded from further analysis.”

Second decision letter

MS ID#: bio.062485R1

MS Title: Glucocorticoids intrinsically redirect naïve CD4+ T cells to the bone marrow for preservation in malnourished mice

Authors: Madeline L. Smith, Jacob Hanes, Takesha R. Foster, Clay Phillips, Kwesi A. Dadzie and Melanie R. Gubbels Bupp

I am happy to tell you that your manuscript has been accepted for publication in Biology Open, pending our standard publication integrity checks. It was accepted on 6th April 2026.

Reviewer 1:

Comments for the author

The revised manuscript is substantially improved and has addressed my major concerns. Overall, this is a careful and interesting study that provides useful insight into how malnutrition and glucocorticoids influence naïve T-cell distribution and survival.

Reviewer 2:

Comments for the author

In the revised version of their manuscript, the authors have carefully addressed my comments. Their conclusions are well supported by their new data or by specific text additions. The discussion of the research is careful and appropriate, and the work is described well, with a good level of detail in the methods. The manuscript is well-written and logical. I have no point to raise. Congratulations to the authors for this nice piece of work.

Reviewer's Responses to Questions

Experimental quality

Does each figure have the proper controls?

If 'No', please indicate reasons in Comments for Author box below.

Reviewer #1:

- Yes

Reviewer #2:

- Yes

Were the data analyzed using appropriate statistical tests?

If 'No', please indicate reasons in Comments for Author box below.

Reviewer #1:

- Yes

Reviewer #2:

- Yes

Reproducibility

Were experiments performed using adequate number of biological replicates?

If 'No', please indicate reasons in Comments for Author box below.

Reviewer #1:

- Yes

Reviewer #2:

- Yes

Does the methods section provide sufficient detail to permit reproducibility?

If 'No', please indicate reasons in Comments for Author box below.

Reviewer #1:

- Yes

Reviewer #2:

- Yes

Completeness

Are the manuscript's conclusions supported by the data?

If 'No', please indicate reasons in Comments for Author box below.

Reviewer #1:

- Yes

Reviewer #2:

- Yes

Scholarship

Do the authors cite and discuss the merits of data that would argue for and against their conclusion?

If 'No', please indicate reasons in Comments for Author box below.

Reviewer #1:

- Yes

Reviewer #2:

- Yes

Does the manuscript title & abstract accurately reflect the contents of the manuscript, without hyperbole?

If 'No', please indicate reasons in Comments for Author box below.

Reviewer #1:

- Yes

Reviewer #2:

- Yes